# A Theoretical Framework for Inference and Learning in Predictive Coding Networks

**Beren Millidge**
MRC Brain Network Dynamics Unit
University of Oxford, UK
`beren@millidge.name`

**Yuhang Song** *
MRC Brain Network Dynamics Unit
University of Oxford, UK
`yuhang.song@ndcn.ox.ac.uk`

**Tommaso Salvatori**
Institute of Logic and Computation
TU Wien, Austria
`tommaso.salvatori@tuwien.ac.at`

**Thomas Lukasiewicz**
TU Wien, Austria
University of Oxford, UK
`thomas.lukasiewicz@tuwien.ac.at`

**Rafal Bogacz**
MRC Brain Network Dynamics Unit
University of Oxford, UK
`rafal.bogacz@ndcn.ox.ac.uk`

## Abstract

Predictive coding (PC) is an influential theory in computational neuroscience, which argues that the cortex forms unsupervised world models by implementing a hierarchical process of prediction error minimization. PC networks (PCNs) are trained in two phases. First, neural activities are updated to optimize the network's response to external stimuli. Second, synaptic weights are updated to consolidate this change in activity — an algorithm called *prospective configuration*. While previous work has shown how in various limits, PCNs can be found to approximate backpropagation (BP), recent work has demonstrated that PCNs operating in this standard regime, which does not approximate BP, nevertheless obtain competitive training and generalization performance to BP-trained networks while outperforming them on various tasks. However, little is understood theoretically about the properties and dynamics of PCNs in this regime. In this paper, we provide a comprehensive theoretical analysis of the properties of PCNs trained with prospective configuration. We first derive analytical results concerning the inference equilibrium for PCNs and a previously unknown close connection relationship to target propagation (TP). Secondly, we provide a theoretical analysis of learning in PCNs as a variant of generalized expectation-maximization and use that to prove the convergence of PCNs to critical points of the BP loss function, thus showing that deep PCNs can, in theory, achieve the same generalization performance as BP, while maintaining their unique advantages.

## 1 Introduction

Predictive coding (PC) is an influential theory in theoretical neuroscience (Mumford, 1992; Rao & Ballard, 1999; Friston, 2003; 2005), which is often presented as a potential unifying theory of cortical function (Friston, 2003; 2008; 2010; Clark, 2015b; Hohwy et al., 2008). PC argues that the brain is fundamentally a hierarchical prediction-error-minimizing system that learns a general world model by predicting sensory inputs. Computationally, one way the theory of PC can be instantiated is with PC networks (PCNs), which are heavily inspired by and can be compared to artificial neural networks (ANNs) on various machine learning tasks (Lotter et al., 2016; Whittington & Bogacz, 2017; Millidge et al., 2020a;b; Song et al., 2020; Millidge et al., 2022). Like ANNs, PCNs are networks of neural activities and synaptic weights. Unlike ANNs, in PCNs, training proceeds by clamping the input and output of the network to the training data and correct targets, respectively, and first letting the neural activities update towards the configuration that minimizes the sum of prediction errors throughout the network. Once the neural activities have reached an equilibrium, the synaptic weights can be

---

*Corresponding author.

updated with a local and Hebbian update rule, which consolidates this change in neural activities. This learning algorithm is called *prospective configuration* (Song et al., 2022), since the activity updates appear to be *prospective* in that they move towards the values that each neuron *should have* in order to correctly classify the input.

Previous work has shown how, in certain limits such as when the influence of feedback information is small or in the first step of inference, PC can approximate backpropagation (BP) and that this approximation is close enough to enable the training of large-scale networks with the same performance as BP (Whittington & Bogacz, 2017; Millidge et al., 2020a; Song et al., 2020); see Appendix A.2 for a full review and comparison. Recent work (Song et al., 2022) has also shown that PCNs trained with prospective configuration under standard conditions can also obtain a training and generalization performance equivalent to BP, with advantages in online, few-shot, and continual learning. Intuitively, the difference between learning in BP and PCNs is that in BP, the error is computed at the output layer and sequentially transported backwards through the network. In PCNs, there is first an inference phase in which the error is redistributed throughout the network until it converges to an equilibrium. Then, the weights are updated to minimize the local errors computed at this equilibrium. Crucially, this equilibrium is in some sense *prospective*: due to its iterative nature, it can utilize information about the activities of other layers that is not available to BP, and this information can speed up the training by not making redundant updates.

However, while the convergence properties of stochastic gradient descent combined with BP are well understood, there is no equivalent understanding of the theoretical properties of prospective configuration yet. In this work, we provide the first comprehensive theoretical study of both the properties of the inference and the learning phase in PCNs. We investigate the nature of the equilibrium computed by the inference phase and analyze its properties in a linear PCN, where an expression for the inference equilibrium can be derived analytically. We show that this equilibrium can be interpreted as an average of the feedforward pass values of the network and the local targets computed by target propagation (TP). We show that the same intuition also holds for nonlinear networks, where the inference equilibrium cannot be computed analytically. Furthermore, we investigate the nature of learning in such networks, which diverges from BP. We present a novel interpretation of PCNs as implementing generalized expectation maximization (EM) with a constrained expectation step, which complements its usual interpretation as variational inference (Friston, 2005; Bogacz, 2017; Buckley et al., 2017). Moreover, we present a unified theoretical framework, which allows us to understand previously disparate results linking PC and BP, to make precise the connection between PC, TP, and BP, and crucially to prove that PCNs trained with prospective configuration will converge to the same minima as BP. We thus show that, in theory, PCNs have at least an equal learning capacity and scalability as current machine learning architectures.

## 2 BACKGROUND AND NOTATION

**Predictive coding networks (PCNs):** Although PCNs can be defined for arbitrary computation graphs (Millidge et al., 2020a; Salvatori et al., 2022a;b), in this paper we assume an MLP structure. A PCN consists of a hierarchical set of layers with neural activations $\{x_0, \ldots, x_L\}$, where $x_l \in \mathcal{R}^{w_l}$ is a real vector of length $w_l$, which is the number of neurons in the layer. In this paper we refer to the input $x_0$ and output $x_L$ layers as the input and output layers respectively. Each layer makes a prediction of the activity of the next layer $\hat{x}_l = f(W_l x_{l-1})$, where $W_l \in \mathcal{R}^{w_l} \times \mathcal{R}^{w_{l-1}}$ is a weight matrix of trainable parameters, and $f$ is an elementwise activation function. During testing, the first layer neural activities are fixed to an input data point $x_0 = d$, and predictions are propagated through the network to produce an output $\hat{x}_L$. During training, both the first and last layers of the network are clamped to the input data point $x_0 = d$ and a target value $x_L = T$, respectively. Both the neural activity and weight dynamics of the PCN can be derived as a gradient descent on a free-energy functional equal to the sum of the squared prediction errors $\mathcal{F} = \sum_{l=0}^{L} \epsilon_l^2$, where $\epsilon_l = x_l - f(W_l x_{l-1})$. Typically, the activity updates are run to convergence $x^*$ (called the inference phase), and the weights are updated once at convergence (called the learning phase):

$$\text{Inference:} \quad \dot{x}_l = -\partial\mathcal{F}/\partial x_l = \begin{cases} -\epsilon_l + W_{l+1}^T \epsilon_{l+1} \cdot f'(W_{l+1}x_l), & \text{for } l < L, \\ -\epsilon_l, & \text{for } l = L, \end{cases} \tag{1}$$

$$\text{Learning:} \quad \dot{W}_l = -(\partial\mathcal{F}/\partial W_l)|_{x_l = x_l^*} = \epsilon_l \cdot f'(W_l x_{l-1}^*) {x_{l-1}^*}^T. \tag{2}$$

In the above equations $\cdot$ denotes elementwise multiplication. The network energy function $\mathcal{F}$ presented here can be derived as a variational free energy under a hierarchical Gaussian generative

model, thus resulting in PCNs having an interpretation as principled approximate inference. For a more detailed description of PCNs, BP, and TP, see Appendix A.1.

**Target propagation (TP)** is an alternative algorithm for training deep neural networks (Lee et al., 2015; Bengio & Fischer, 2015). Its core intuition is that the ideal activity for a layer is *the activity that, when passed through the forward pass of the network, outputs the desired label or target exactly*. If we can compute this quantity (which we call a local *target*), then we can update the feedforward weights to minimize the difference between the actual activity and the local target, and so ensure that eventually the feedforward pass of the network will compute the correct targets. Computing the local targets involves computing the inverse of a network. For instance, given a forward pass through a layer $x_l = f(W_l x_{l-1})$, the local target $t_l$ can be defined in terms of the local target of the layer above as $t_l = W_{l+1}^{-1} f^{-1}(t_{l+1})$. Given this local target, we can define the local prediction error as $\epsilon_l^{\text{TP}} = t_l - x_l$, which is minimized by weight updates. In practice, since explicit inverses are expensive to compute, and most neural networks are not invertible, the pseudoinverse is used or approximated(Lee et al., 2015; Bengio, 2020). The TP recursion is identical to the BP recursion except that it uses inverses instead of derivatives and transposes. Although BP and TP seem very different, a key result that we show is that the inference phase of a PCN effectively provides an interpolation between the BP and TP targets.

## 3 RESULTS

### 3.1 INFERENCE IN PCNS

To build a comprehensive mathematical theory of PCNs, we must deeply understand both inference and learning. We first focus on the former, and explore the connection between the inference equilibrium reached in a PCN and both BP and TP. We show that the PCN inference equilibrium interpolates between BP and TP, and that the weight updates approximate BP when the feedforward influence of the data on the network is large, and so the inference equilibrium is close to the feedforward pass values, while it approximates TP when the feedback influence of the target is large.

First, we recapitulate the result that PC approximates BP when the feedforward influence of the data predominates, and thus the equilibrium activities are close to their feedforward pass values.

**Theorem 3.1** *Let $\mathcal{M}$ be a PCN with neural activities $\{x_0, \ldots, x_L\}$ and prediction errors $\{\epsilon_1, \ldots, \epsilon_L\}$, where the final prediction error equals the gradient of the loss relative to the output ($\epsilon_L = \partial \mathcal{L}/\partial x_L$). Then, the prediction error at equilibrium at a layer $\epsilon_l^*$ approximates the BP gradient $\partial \mathcal{L}/\partial x_l$ up to order $\mathcal{O}(\epsilon_f)$ and hence converges to exact BP as $x_l^* \to \bar{x}_l$, where $\epsilon_f = f'(W_{l+1} x_l^*) - f'(W_{l+1} \bar{x}_l)$, with $f'$ being the activation function derivative and $\bar{x}_l$ the feedforward pass value at layer $l$.*

**Proof.** Consider the equilibrium of the activity dynamics of a layer in terms of the prediction errors: $\dot{x}_l = \epsilon_{l+1} \cdot f'(W_{l+1} x_l) W_{l+1}^T = 0 \implies \epsilon_l^* = \epsilon_{l+1}^* \cdot f'(W_{l+1} x_l^*) W_{l+1}^T = \epsilon_{l+1}^* (\partial \hat{x}_{l+1}/\partial x_l)|_{x_l = x_l^*}$. This recursive relationship is identical to that of the BP gradient recursion $\partial \mathcal{L}/\partial x_l = \partial \mathcal{L}/\partial x_{l+1} \partial x_{l+1}/\partial x_l$ except that the gradients are taken at the activity equilibrium instead of the feedforward pass values. To make the difference explicit, we can add and subtract the feedforward pass values to obtain

$$
\begin{aligned}
\epsilon_l^* &= \epsilon_{l+1}^* \partial f(W_{l+1} \bar{x}_l)/\partial \bar{x}_l + \epsilon_{l+1}^* \cdot [f'(W_{l+1} x_l^*) - f'(W_{l+1} \bar{x}_l)] W_{l+1}^T \\
&= \underbrace{\epsilon_{l+1}^* \partial f(W_{l+1} \bar{x}_l)/\partial \bar{x}_l}_{\text{BP}} + \mathcal{O}(\epsilon_{l+1}^* \epsilon_f).
\end{aligned} \tag{3}
$$

This linear dependence of the approximation error to BP on the difference between $f'(W_{l+1} x_l^*)$ and $f'(W_{l+1} \bar{x}_l)$ clearly implies that PC converges to BP as $x_l^* \to \bar{x}_l$.

In fact, we obtain additional more subtle conditions. The above proof also implies convergence to BP when the feedback error $\epsilon_{l+1} \to 0$ and additionally when the activation function derivatives $f'$ are the same even when the equilibrium activities are not the same as the feedforward pass values. Overall, the feedforward influence on the PCN has thus the effect of drawing the equilibrium activity close to the feedforward pass, resulting in the PCN approximating BP. This result is closely related to previous results relating the PCN dynamics to BP (Whittington & Bogacz, 2017; Millidge et al., 2020a; Song et al., 2020; Salvatori et al., 2022b), reviewed in Appendix 6.

We next consider the second limiting case where there is only feedback influence on the PCN by considering the 'input-unconstrained' case where the input layer of the PCN is not fixed to any data.

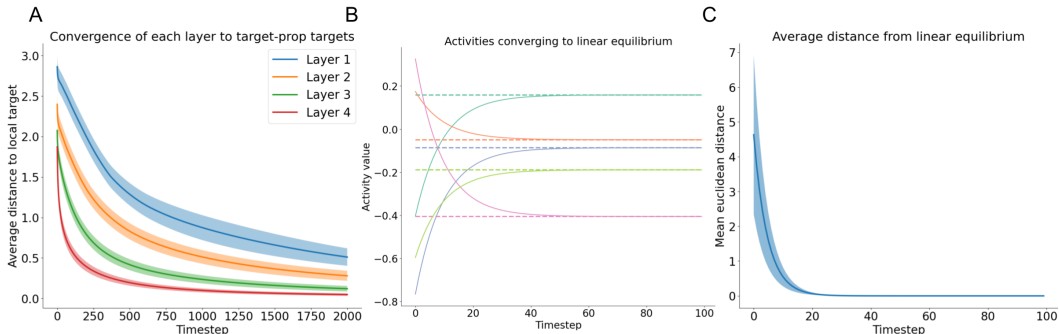

Figure 1: **A**: Evolution of the activities converging towards the TP targets during an inference phase in a nonlinear input-unconstrained network, averaged over 50 seeds. **B** Example convergence of activities in a layer towards the linear equilibrium computed in Theorem 3.3 in a 3-layer linear PCN. **C** Total Euclidean distance of all activities of a 3-layer PCN from the linear equilibrium during inference, averaged over 50 seeds. Error bars represent standard deviations.

**Theorem 3.2** *Let $\mathcal{M}$ be a PCN with neural activities $\{x_0, \ldots, x_L\}$ with an invertible activation function $f$ and square invertible weight matrices $\{W_1, \ldots, W_L\}$. Let the output layer be clamped to a target $x_L = T$ and the input layer be unclamped. Then, the equilibrium activities $x_l^*$ of each layer are equal to the TP targets of that layer.*

**Proof.** If the input layer is unconstrained, and the network is invertible, then there is a unique global minimum where $\epsilon_l = 0$ for all $l$. Substitute in the TP target definition for $x_l$. We have that $\epsilon_{l+1} = x_{l+1} - f(W_{l+1}x_l) = x_{l+1} - f(W_{l+1}(W_{l+1}^{-1}f^{-1}(x_{l+1}))) = x_{l+1} - x_{l+1} = 0$. This holds for all $\epsilon_l$. To see why this minimum is unique, consider the prediction error of the final layer, which is clamped to the target $\epsilon_L = T - f(W_L x_{L-1}) = 0$. By the assumption of invertible $f$ and $W_L$, this equation has only one solution $x_{L-1} = W_L^{-1}f^{-1}(T)$. This applies recursively down to each layer of the network substituting in $x_{L-1}$ for $T$. If the invertibility assumption is relaxed, as in realistic neural networks, there may be multiple equally valid targets and the input-unconstrained PCN will converge to one such solution. A more intuitive dynamical proof is given in Appendix A.6.3. In Figure 1 panel A, we demonstrate this empirically by showing that for an input-unconstrained 5-layer nonlinear network with *tanh* activation functions, the layer activities reliably towards the local targets. Interestingly, this occurs in a distinctly layerwise fashion with layers closer to the output converging first, highlighting the temporal dynamics of information transmission through the network. Full details for all networks used in the experiments in can be found in Appendix A.10.

However, during PCN training, the input layer is clamped to the data while the output layer is clamped to the target. While it is not possible to derive an analytical expression for the equilibrium for a general nonlinear network, we can derive an analytical formula for the equilibrium in the linear case.

**Theorem 3.3** *Let $\mathcal{M}$ be a linear PCN with neural activities $\{x_0, \ldots, x_L\}$ and prediction function $\hat{x}_l = W_l x_{l-1}$, then the equilibrium value of a layer $x_l^*$ is computable in terms of the equilibrium values of the layers below $x_{l-1}^*$ and above $x_{l+1}^*$ as $x_l^* = (I + W_{l+1}^T W_{l+1})^{-1}[W_l x_{l-1}^* + W_{l+1}^T x_{l+1}^*]$*

**Proof**. By setting $\dot{x}_l = 0$, we obtain

$$\begin{aligned}
\dot{x}_l &= -\epsilon_l + W_{l+1}^T \epsilon_{l+1} = -x_l + W_l x_{l-1} + W_{l+1}^T[x_{l+1} - W_{l+1}x_l] = 0 \\
&= -(I + W_{l+1}^T W_{l+1})x_l + W_{l+1}^T x_{l+1} + W_l x_{l-1} = 0 \\
&\implies x_l^* = (I + W_{l+1}^T W_{l+1})^{-1}[W_l x_{l-1}^* + W_{l+1}^T x_{l+1}^*].
\end{aligned} \tag{4}$$

We observe that the linear equilibrium consists of a sum of a 'feedforward' term $W_l x_{l-1}$ and a 'feedback' term $W_{l+1}^T x_{l+1}$, both weighted by a prefactor $(I + W_{l+1}^T W_{l+1})$, which effectively measures the degree of orthogonality of the forward pass weight matrix. This expression neatly represents the 'prospective equilibrium' for the activities of a PCN in terms of a feedforward component, which we can think of as trying to pull the equilibrium state of the network towards the initial feedforward pass values, and a feedback component, which we can interpret as trying to pull the equilibrium activity towards the TP targets. Also, note that in the linear case when all prediction errors are zero ($\epsilon_l^* = 0$), the feedforward and feedback influences are exactly equal $W_l x_{l-1}^* = W_{l+1}^{-1} x_{l+1}^*$ (proof in Appendix A.6.2.) which implies that at the global optimum of the PCN (if it is reachable), the feedforward pass activies are exactly equal to the TP targets. Additional mathematical results

Figure 2: **Left:** PC interpolates between BP and TP depending on the precision ratio. **Right:** Intuitions are maintained in the nonlinear case. **Right**: The cosine similarity between prediction errors and BP gradients as well as neural activities and TP targets during the inference phase for a 3-layer nonlinear (tanh) network. As predicted by our linear theory, the similarity to targetprop targets increases during inference while the similarity to the backprop gradient decreases.

demonstrating the convexity of the optimization problem in the linear case, and deriving explicit differential equations for the activity dynamics are given in Appendices A.4 and A.5. In Figure 1 panel B, we showcase the node values in a 5-node PCN converging to the exact linear equilibrium predicted by the theory. Moreover, in panel C, we plot the mean euclidean distance of all neurons from the predicted equilibrium values as a function of inference step for a 3-layer linear network which demonstrate rapid and robust convergence. While these results have been primarily shown for the linear case, we also wish to understand the extent to which they hold empirically in the nonlinear case. In Figure 2B, we plot the similarities of the equilibrium activities of a nonlinear 3-layer PCN with a `tanh` activation function to both targetprop targets and backprop gradients and show that the same intuitions hold in the nonlinear case as predicted by our linear theory.

## 3.2 INFERENCE IN PRECISION-WEIGHTED PCNS

We have seen that the equilibrium activities are determined by a sum of feedforward and feedback influences. We demonstrate that the analytical precision equilibrium here is correct empirically in Figure 3A and B which shows that linear PCNs rapidly and robustly converge to the analytical solution we derive. However, we have not yet defined a way to change the relative weighting of these influences. The Gaussian generative model that underlies the derivation of the PCN does, however, give us precisely such a mechanism by utilizing the inverse variance or *precision* parameters of the Gaussian distribution $\Pi_l = \Sigma_l^{-1}$, which thus far we have tacitly assumed to be identity matrices. If we relax this assumption, we obtain a new precision-weighted dynamics for the activities:

$$\dot{x}_l = \Pi_l \epsilon_l - \Pi_{l+1} \epsilon_{l+1} \cdot f'(W_{l+1} x_l) W_{l+1}^T. \tag{5}$$

From these dynamics, we can derive an analytical expression for the precision-weighted equilibrium.

**Theorem 3.4** *Let $\mathcal{M}$ be a linear PCN with layerwise precisions $\{\Pi_1, \ldots, \Pi_L\}$. Then, the precision-weighted equilibrium value of a layer $x_l^*$ is computable in terms of the equilibrium values of the layers below $x_{l-1}^*$ and above $x_{l+1}^*$ as $x_l^* = (I + \Pi_l^{-1} W_{l+1}^T \Pi_{l+1} W_{l+1})^{-1} [W_l x_{l-1}^* + \Pi_l^{-1} W_{l+1}^T \Pi_{l+1} x_{l+1}^*]$.*

A proof is given in Appendix A.6.1. The result is similar to the previous linear equilibrium and identical to it when $\Pi_l = \Pi_{l+1} = I$. Heuristically, or exactly with diagonal precisions, this expression lets us think of the precision *ratios* $\Pi_{l+1}/\Pi_l$ as determining the relative weighting of the feedforward and feedback influences on the network dynamics. These allow us to derive more precise approximate statements about the relative weighting of the feedforward and feedback influences leading to convergence either to the feedforward pass values or the TP targets. For instance, suppose the feedback precision ratio is large, so that in the prefactor the term $\Pi_l^{-1} W_{l+1}^T \Pi_{l+1} W_{l+1}$ dominates. This will occur when the precision at the next layer is substantially larger than that at the current layer, i.e., the prediction errors at the next layer are 'upweighted'. In this case, for the prefactor, we have that $(I + \Pi_l^{-1} W_{l+1}^T \Pi_{l+1} W_{l+1})^{-1} \approx (\Pi_l^{-1} W_{l+1}^T \Pi_{l+1} W_{l+1})^{-1} \approx W_{l+1}^{-1} \Pi_{l+1}^{-1} W_{l+1}^{-T} \Pi_l$, since the identity term in the prefactor becomes negligible. As such, we can approximate the equilibrium as

$$x_l^* \approx (W_{l+1}^{-1} \Pi_{l+1}^{-1} W_{l+1}^{-T} \Pi_l) [W_l x_{l-1}^* + \Pi_l^{-1} W_{l+1}^T \Pi_{l+1} x_{l+1}^*]$$
$$\approx W_{l+1}^{-1} \Pi_{l+1}^{-1} W_{l+1}^{-T} \Pi_l W_l x_{l-1}^* + W_{l+1}^{-1} x_{l+1}^* \approx W_{l+1}^{-1} x_{l+1}^*. \tag{6}$$

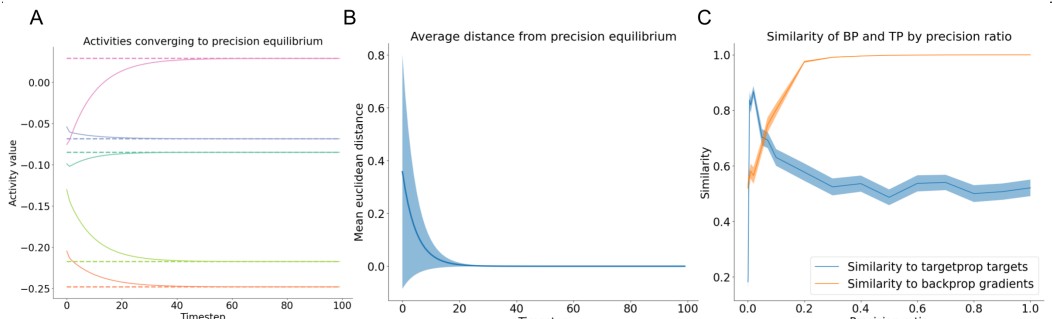

Figure 3: **A** Example activities of neurons in a layer converging to the analytical precision equilibrium during an inference phase in a 3-layer linear PCN. **B** Total Euclidean distance from the analytically computed precision equilibrium averaged over 50 random initializations. **C** The similarity between the equilibrium activities and the targetprop targets, as well as the equilibrium prediction errors and the backprop gradients at the end of inference in a nonlinear 3-layer tanh PCN for different precision ratios $\frac{\Pi_l}{\Pi_{l+1}}$. We observe the same qualitative effects of increasing similarity to backprop with high bottom-up precision and increasing similarity to target-prop with high top-down precision in the nonlinear case as predicted by our linear theory. Error bars are standard deviation over 50 seeds.

Here, we can discount the first term: as $\Pi_l^{-1} W_{l+1}^T \Pi_{l+1} W_{l+1}$ is assumed to be large, its inverse $W_{l+1}^{-1} \Pi_{l+1}^{-1} W_{l+1}^{-T} \Pi_l$ is small. Effectively, if the relative ratio of feedback to feedforward precision is large, then the balance between the feedforward and feedback stream at equilibrium is tilted towards the feedback stream, and the activities at equilibrium approximate the TP targets. Conversely, if the precision ratio is low, then the $(\Pi_l^{-1} W_{l+1}^T \Pi_{l+1} W_{l+1})$ term in the prefactor is very small. In this case, we can approximate the prefactor simply by the identity matrix as $(I + \Pi_l^{-1} W_{l+1}^T \Pi_{l+1} W_{l+1})^{-1} \approx I^{-1} \approx I$. This then lets us write the equilibrium as

$$x^* = W_l x_{l-1}^* + \Pi_l^{-1} W_{l+1}^T \Pi_{l+1} x_{l+1}^* \approx W_l x_{l-1}^*, \qquad (7)$$

where the approximation assumes that the ratio of feedback to feedforward precision is small, i.e., $\Pi_l^{-1} \Pi_{l+1} \approx 0$. That is, the activities at equilibrium approximately equal the feedforward pass values. While this analysis only applies to linear networks, we show in Figure 3C that a similar qualitative result applies even to nonlinear networks where the similarity to the backprop gradients increases with the ratio of bottom-up to top-down precision and vice-versa for the similarity to the targetprop targets, thus showing that the intuitions gained in our analysis generalize beyond the linear setting.

### 3.3 LEARNING IN PCNs

We now present a theoretical framework for understanding the learning phase of PCNs and its convergence properties. Our key result is that the learning algorithm should ultimately converge to a critical point of the BP loss $\mathcal{L}$. To do so, we need to develop an alternative mathematical interpretation of PCNs as an expectation-maximization (EM) algorithm (Dempster et al., 1977), which complements the standard view of PC as variational inference. The ultimate goal of PC is to be able to achieve *local learning* such that the weights of the network can be updated to minimize only a locally defined loss using local information, but which nevertheless will ultimately result in minimizing the global (output) loss of the network. One intuition as to how to achieve this is to compute local 'targets' for each layer, where the ideal target is the local activity value that would minimize the output loss of the network. If we had such targets, then local learning would be simple — we would simply update the weights to minimize the difference between the activity of the layer and its local target. Since there would be a local target for each layer, all information would be available locally.

The fundamental problem is that we do not know the correct local targets. However, given the input data value and the label, we can *infer* the correct local targets at each layer. Thus, the computation of the optimal targets is a Bayesian inference problem. An alternative way to think about this is as a missing data problem. Ideally, we would like to minimize the loss with the optimal local targets and the input data. However, unfortunately, the local targets are 'missing' and so to proceed with our optimization of the weights, we must infer them first. This is the classic justification for the EM-algorithm (Dempster et al., 1977; Minka, 1998), which involves first an 'expectation' (E) step, which infers the posterior distribution over the missing data, and then a 'maximization' step, which maximizes the log likelihood of the parameters averaged over the posterior. PCNs do not infer the full posterior distribution but only a delta distribution around the most likely value of the targets.

To mathematically formalize this, suppose that we have a discriminative PCN with input data $D$, output target $T$, and estimated local targets $\{x_l\}$. We want to find the most likely values $x_l$ for the local targets. This can be expressed as a maximum a-posteriori (MAP) inference problem:

$$\{x_l^*\} = argmax_{\{x_l\}} \, p(\{x_l\}|T, D, W) = argmax_{\{x_l\}} \, \ln p(T|\{x_l\}, D, W) + \ln p(\{x_l\}|D, W). \quad (8)$$

Given a set of $x_l^*$, we can then interpret the learning phase as maximizing the log likelihood of the targets with respect to the weight parameters averaged around the posterior distribution of the local targets, which we have approximated with its MAP value $p(\{x_l\}|T, D, W) \approx \prod_{l=1}^{L} \delta(x_l - x_l^*)$. That is, we can write out the PC equivalently as an EM algorithm:

$$\text{E-step: } argmax_{\{x_l\}} \, p(\{x_l\}|T, D, W), \quad (9)$$

$$\text{M-step: } argmax_W \, \mathbb{E}_{p(\{x_l^*\}|T,D,W)}[p(T|\{x_l^*\}, D, W)]. \quad (10)$$

These dual interpretations of PC as both EM and variational inference are not surprising. As shown in Neal & Hinton (1998), any EM algorithm can be expressed as a dual descent on a variational free energy functional $\mathcal{F}$. By the known convergence properties of the EM algorithm, we can derive that the PCN converges to a minimum of the variational free energy $\mathcal{F}$.

### 3.4 Convergence of PC to Critical Points of the Loss

A-priori, the results of Section 3.3 tell us little about the extent to which the minima of $\mathcal{F}$ correspond to useful minima of the BP, or supervised loss $\mathcal{L}$. Next, we address this question by re-interpreting PC from performing EM on the variational free energy $\mathcal{F}$ to performing constrained EM on the BP loss $\mathcal{L}$, and hence showing that PCNs are guaranteed to converge to a minimum of $\mathcal{L}$. This means that PC possesses the same convergence properties that BP enjoys, and thus that, in theory, PCNs possess the same learning capabilities as BP. This ultimately implies that the PC learning rule, although different from BP, can be scaled to train large-scale deep networks with the same performance as BP.

First, we want to understand how the variational free energy $\mathcal{F}$ and the BP loss $\mathcal{L}$ are related. Recall that the variational free energy in PCNs is just a sum of the local energies in each layer, thus we can define $\mathcal{F} = \sum_l E_l$, where the 'local energy' is defined as $E_l = \epsilon_l^2 = (x_l - f(W_l x_{l-1}))^2$. Secondly, in PCNs performing classification, the BP loss $\mathcal{L}$ is just the loss at the output layer in PCNs, i.e., $\mathcal{L} \equiv E_L$ (while the BP loss also depends on all the other weights on the network, the same is true for the output layer of the PCN where this dependence is made implicit through the inference process). This allows us to express the variational free energy as a sum of the BP loss and a remaining 'residual loss' $\tilde{E}$ of the rest of the network:

$$\mathcal{F} = \mathcal{L} + \tilde{E}, \quad (11)$$

where the residual energy is defined as $\tilde{E} = \sum_{l=1}^{L-1} E_l$. Now, we want to understand how to relate general critical points of $\mathcal{F}$ to critical points of $\mathcal{L}$. The key will be to show that we can reinterpret the dynamics of PCN as performing a constrained EM algorithm on $\mathcal{L}$ and not $\mathcal{F}$ and thus by the known convergence of EM (Neal & Hinton, 1998), the PCN is guaranteed to converge to a critical point of $\mathcal{L}$. We first show that the *marginal condition* holds.

**Lemma 3.1** *The marginal condition: At equilibrium, the gradients of the BP loss relative to the activities are exactly equal and opposite to the gradients of the residual loss:* $\partial \mathcal{L}/\partial x_l = -\partial \tilde{E}/\partial x_l$.

A proof is given in Appendix A.7.2. Intuitively, the marginal condition requires that at equilibrium, any decrease of $\mathcal{L}$ must be exactly offset by an increase in $\tilde{E}$ or vice versa, since if this were not the case, then one could decrease $\mathcal{F}$ by moving in a direction that decreased $\mathcal{L}$ more than $\tilde{E}$ increased or vice versa, which would imply that we were not in an optimum. To gain an intuition for how this works in practice, in Figure 4A, we plot the values of these energies in a randomly initialized 4-layer relu network as they evolve throughout the inference phase. If initialized in the correct way, the total free energy and backprop loss will decrease during inference while the residual loss will increase until an equilibrium, given by the marginal condition, is reached. We can now state our main result.

**Theorem 3.5** *Let $\mathcal{M}$ be a PCN that minimizes a free energy $\mathcal{F} = \mathcal{L} + \tilde{E}$. Given a sufficiently small learning rate and that for each minibatch activities are initialized subject to the energy gradient bound: $-\partial \mathcal{L}/\partial x_l^T \partial \tilde{E}/\partial x_l \leq ||\partial \mathcal{L}/\partial x_l||^2$ (derived in Appendix A.7.3) then, during training, the PCN will converge to a critical point of the BP loss $\mathcal{L}$.*

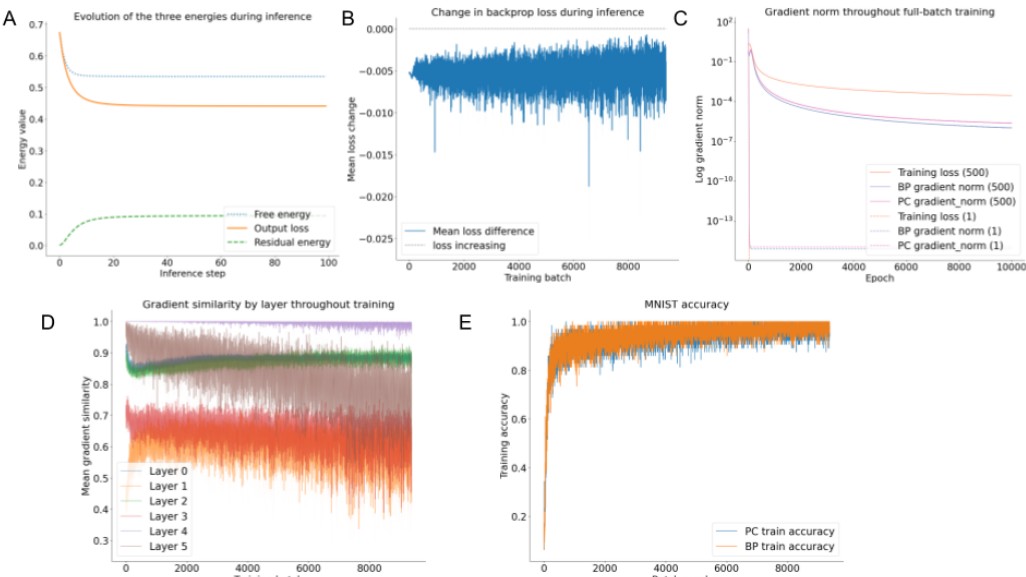

Figure 4: **A**: Evolution of the decomposed energies during an inference phase of a 4-layer relu network. The output loss $\mathcal{L}$ continuously decreases throughout the inference phase, while the residual energy $\tilde{E}$ increases. **B**: To verify the energy gradient bound holds in practice with a standard step-size of $0.1$, we plot the change in backprop loss between the beginning and end of the inference phase throughout training. We find that the change is always negative (loss decreasing). **C**: To verify convergence to a minimum of the backprop loss, we plot the training loss and norm of the BP and PC gradients during full-batch training of either 500 or 1 MNIST digits. When the dataset comprises a single digit, the PCN minimizes training loss to 0 with gradient norms of order $10^{-16}$ or within numerical precision given finite learning rate, exactly as predicted by the theory. For $500$ MNIST digits a very low loss and gradient value of order $10^{-6}$ is achieved but then training stalls, perhaps due to the finite learning rate. **D**: Similarity scores for each layer in a 6 layer network throughout training between the PC and BP weight updates. Even though PC updates are very different from BP, they still empirically converge to the same minima. This difference allows PC to exhibit different and potentially superior optimization performance. **E**: Accuracy of PC and BP on MNIST dataset showing that both learn at a similar rate and reach equivalent performance.

The proof proceeds in several steps. We first express the inference and learning phases as constrained minimizations of $\mathcal{L}$ and not $\mathcal{F}$. We use this to write the PCN dynamics as an EM algorithm on $\mathcal{L}$ and then apply the known convergence results of EM to prove convergence to a critical point of $\mathcal{L}$.

Recall that we assumed an initialization that satisfies the energy gradient bound $-\partial\mathcal{L}/\partial x_l^T \partial\tilde{E}/\partial x_l < ||\partial\mathcal{L}/\partial x_l|||^2$. We prove in Appendix A.7.3 that under such conditions the supervised loss is guaranteed to decrease during inference. This is because initializing with the energy bound condition implies that the gradient of the backprop loss is greater than the negative gradient of the residual energy. If the dynamics of the gradients of the backprop loss and residual energy are continuous (which technically requires continuous time inference or an infinitesimal learning rate), then by the intermediate value theorem, for the gradients to 'cross' such that the residual energy gradient is greater than the backprop loss gradient, and hence for the loss to increase, they must first attain a point at which they are equal. This point is given by the marginal condition and is the inference equilibrium, and hence the inference procedure terminates at this condition, thus the backprop loss cannot increase during inference. Intuitively, the dynamics of inference in these conditions will simply 'roll down the slope' of continuing to decrease $\mathcal{L}$ until an equilibrium given by marginal condition is reached. With a small enough learning rate, there should never be an 'overshoot' where the supervised loss increases. While this argument technically requires an infinitesimal learning rate, in Figure 4B, we plot the change of backprop loss across the inference phase throughout training for a deep nonlinear network and find it is always negative, even with a step-size of $0.1$, implying that this condition seems robust to finite step-sizes in practice. Thus, we can interpret the dynamics of inference in another way, namely, a constrained minimization of $\mathcal{L}$

subject to the constraints of the energy gradient bound:

$$x_l^* = \underset{x_l}{argmin} \ L \text{ such that } - \partial\mathcal{L}/\partial x_l^T \partial\tilde{E}/\partial x_l < ||\partial\mathcal{L}/\partial x_l|||^2. \tag{12}$$

Under the assumption that the inference phase always converges to some solution which must satisfy the marginal condition, it is clear to see that the solutions of the two optimization processes are the same. Crucially, this has allowed us to reinterpret the inference phase as a minimization of $\mathcal{L}$ and not $\mathcal{F}$. Intuitively, what we have done is reinterpreted an unconstrained minimization of $\mathcal{F}$, which must find an equilibrium between a trade-off of $\mathcal{L}$ and $\tilde{E}$ as instead a constrained minimization of $\mathcal{L}$ subject to the condition that the trade-off that it implicitly makes with $\tilde{E}$ is not greater than the equilibrium condition allowed by the unconstrained minimization of $\mathcal{F}$. The next step is to show that we can also express the weight update in terms of a minimization of $\mathcal{L}$ and not $\mathcal{F}$. Luckily this is straightforward. Recall from Theorem 3.1 that we can express the prediction errors at equilibrium in terms of a BP-like recursion through the equilibrium values:

$$\epsilon_l^* = \epsilon_{l+1}^* \partial f(W_l x_l^*)/\partial x_l^*. \tag{13}$$

The reason that the gradients computed by PCNs differ from BP is that the BP gradients are evaluated at the feedforward pass values $\bar{x}_l$, while the $\epsilon_l^*$ recursion involves the activities at their equilibrium values $x_l^*$. This means, however, that we can interpret the weight update step in PCNs as performing *BP through the equilibrium values of the network*. This lets us express the weight update of the PCN in terms of a gradient descent on $\mathcal{L}$. Since we can write both the inference and learning phases of the PC network in terms of a minimization of the loss $\mathcal{L}$, we can express it in terms of an EM algorithm:

$$\text{E-step (inference): } x_l^* = \text{argmin}_{x_l} \ \mathcal{L} \text{ such that } - \partial\mathcal{L}/\partial x_l^T \partial\tilde{E}/\partial x_l < ||\partial\mathcal{L}/\partial x_l|||^2 \tag{14}$$

$$\text{M-step (learning): } \text{argmin}_{W_l} \ \mathbb{E}_{p(\{x_l^*\}|y,D,W)}[(\mathcal{L}(\{x_l^*\}, W)], \tag{15}$$

where we can interpret the posterior over $x_l$ as a delta function around the equilibrium value $p(x_l|y, D, W) = \delta(x_l - x_l^*)$. Crucially, since $\mathcal{L}$ is guaranteed to either decrease or stay the same at each E and M step, the algorithm must converge to a local minimum (or saddle point) of $\mathcal{L}$ (Neal & Hinton, 1998). This occurs even though the E step is constrained by the energy gradient bound, since as long as the network is initialized on the correct 'side' of the bound, the minimization is guaranteed to at least decrease $\mathcal{L}$ a small amount during the convergence and cannot increase it unless the marginal condition is violated. To empirically demonstrate convergence to a minimum of the backprop loss, in Figure 4C, we took a simple case where convergence is straightforward – full gradient descent on a trivial dataset of either a 1 or 500 MNIST digits. We plotted the norms of the backprop gradient and full PCN gradient during training. In both cases we find that both the training loss and backprop gradient norm (which should be 0 at the minimum of the backprop loss) decrease rapidly and quickly reach close to 0 within a numerical tolerance. For a single MNIST input, the training loss goes to 0 and the backprop gradient norm stabilizes at approximately $10^{-16}$, indicating full convergence except for numerical errors. For the 500 MNIST digits, the training loss and gradient norms rapidly decrease to the order of $10^{-4}$ and stabilize there, likely due to a finite learning rate size and potential insufficient expressivity to fully fit the batch. Nevertheless, these results demonstrate that PCNs can robustly and rapidly converge very close to a minimum of the backprop loss as predicted by our theory. In Figure 4E, we show that a similar accuracy is obtained by training on the full MNIST dataset between PCNs and BP, also validating our theory. Moreover, this equivalent performance and convergence occurs without the PC and BP gradients being similar. In Figure 4D, we plot the cosine similarity during training of each the PCN weight updates of layer of a 5 layer PCN to the BP updates and find that they remain consistently different across layers and during training. This further supports our main theoretical claim that PCNs exhibit different convergence behaviour to BP, but can nevertheless ultimately minimize the backprop loss function.

## 4 DISCUSSION

In this paper, we have provided a novel theoretical framework for understanding and mathematically characterising both inference and learning in PCNs. We have provided the first explicit mathematical characterisation of the inference equilibrium in the case of linear networks, explored how they vary with the precision parameter that determines the relative weighing of the feedforward and feedback influences on the network, and discovered a novel connection between PCNs and TP such that the inference equilibrium of a PCN lies on a spectrum between the TP and BP solution. Secondly, we have provided a novel mathematical interpretation of PC as a generalized expectation-maximization algorithm where the inference phase can be expressed as a constrained optimization.

ACKNOWLEDEGMENTS

This work has been supported by BBSRC grant BB/S006338/1 and MRC grant MC_UU_00003/1. Thomas Lukasiewicz has been supported by the Alan Turing Institute under the UK EPSRC grant EP/N510129/1 and by the AXA Research Fund.

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

# A APPENDIX

## A.1 INTRODUCTION TO MAIN ALGORITHMS

### A.1.1 PREDICTIVE CODING

While originally motivated by information-theoretic concerns about compressing sensory data and minimizing redundancy in the brain (Mumford, 1992; Rao & Ballard, 1999; Huang & Rao, 2011),

predictive coding can be most elegantly derived as a variational Bayesian inference algorithm (Beal, 2003; Ghahramani et al., 2000). Specifically, variational methods aim to turn an inference problem into an optimization problem by, instead of computing the intractable Bayesian posterior directly, minimizing an upper bound upon the divergence between the approximate posterior and true posterior. Mathematically, we can formalize this process as follows. We are given some observations $o$ and a latent state $x$, which we are trying to infer. The true posterior is $p(x|o) = \frac{p(o,x)}{p(o)}$ where $p(o,x) = p(o|x)p(x)$ is the generative model of the data, and is assumed known. Variational methods approximate the true posterior $p(x|o)$ by beginning with an approximate posterior distribution $q(x|o)$ and then fitting to the true posterior by minimizing the divergence $KL[q(x|o)||p(x|o)]$. By itself, this objective is intractable, since it contains the intractable posterior. However, the key insight is that we can instead minimize an upper bound on this objective called the variational free energy $\mathcal{F}$.

$$\begin{aligned}
\mathcal{F} &= \mathbb{E}_{q(x|o)}\big[\ln q(x|o) - \ln p(o,x)\big] \\
&= KL[q(x|o)||p(x|o)] - \ln p(o) \\
&\geq KL[q(x|o)||p(x|o)]
\end{aligned} \tag{16}$$

If we parametrize this approximate posterior distribution in some way, for instance with parameters $\theta$, which we denote by $q(x|o;\theta)$, then we can write an abstract variational inference algorithm as simply minimizing the free energy,

$$\begin{aligned}
\theta^* &= \underset{\theta}{argmin}\ \mathcal{F}(\theta, o) \\
&= \underset{\theta}{argmin}\ \mathbb{E}_{q(x|o;\theta)}\big[\ln q(x|o;\theta) - \ln p(o,x)\big]
\end{aligned} \tag{17}$$

To turn this specification into a practical algorithm, we must first specify concretely what the approximate posterior and the generative model are, and secondly we must specify a method of minimizing the free energy. To derive the predictive coding algorithm, we first assume a hierarchical set of latent variables $[x_0 \ldots x_L]$ where $L$ is the number of layers in the hierarchy and we set the lowest level to be the input data: $x_0 \equiv D$. We then assume that these layers can be related according to some parametrizable nonlinear function with parameters $\phi$ such that $x_l = f(\phi, x_{l+1})$. From this, if we assume all 'noise' in the system is Gaussian, we can build a probabilistic Gaussian generative model $p(x_0 \ldots x_L) \equiv \prod_{l=1}^{L} p(x_l|x_{l+1}) \equiv \mathcal{N}(x_l; f(x_{l+1}, \phi), \sigma_l)$. Similarly, we can define the approximate posterior to be a Gaussian distribution factorized across layers $q(x_1 \ldots x_L|x_0) \equiv \prod_{l=1}^{L} q(x_l; \theta)$. Then, under a Laplace approximation about the variances of the variational distribution (see Appendix A for a full derivation), the free energy can be considerably simplified to a sum of squared prediction errors (Friston, 2005; Buckley et al., 2017),

$$\mathcal{F}(o, x, \phi) \approx \sum\nolimits_{l=1}^{L} \epsilon_l^2 \tag{18}$$

Where $\epsilon_l = x_l - f(\phi, x_{l+1})$ which can be thought of as the prediction error between the estimated mean of the current layer, and the mean predicted by the mean of the layer above, where this prediction is done using the nonlinear function $f$ parametrized by $\phi$. Finally, to actually minimize this objective, we perform a gradient descent on the free energy with respect to both $x$ and $\phi$, for which we can compute the required gradients as follows,

$$\begin{aligned}
\dot{x} &\equiv -\frac{\partial \mathcal{F}}{\partial x} = -\epsilon_l + \epsilon_{l+1}\frac{\partial f(\phi, x_{l+1})}{\partial x_l} \\
\dot{\phi} &\equiv -\frac{\partial \mathcal{F}}{\partial \phi} = -\epsilon_l\frac{\partial f(\phi, x_l)}{\partial \phi}
\end{aligned} \tag{19}$$

In practice, we typically optimize the $x$s first to convergence, a process we call 'inference' and at convergence we run a single step of optimization of the $\phi$ parameters, which we call 'learning'. This can be justified from a probabilistic standpoint as the $x$ parameters are parameters of the *variational density* $q(x|o;x)$ while the $\phi$ parameters are part of the *generative model* $p(o,x;\phi)$. If we specialize predictive coding further, to the case of a perceptron like artificial neural network where we have that $x_l = f(W_l x_{l+1})$ where $f$ is an elementwise activation function such as a tanh or a relu nonlinearity, and we split out the parameters into layerwise weight matrices $\phi = \{W_l\}$, we can write the learning rules as follows.

$$\begin{aligned}
\dot{x}_l &= -\epsilon_l + \epsilon_{l+1}f'(W_l x_l)W_l^T \\
\dot{W}_l &= -\epsilon_l f'(W_l x_l)x_l^T
\end{aligned} \tag{20}$$

Where $f'$ denotes the partial derivative of the activation function. Due to the simplicity, and relative biological plausibility of its update rules, predictive coding networks have often considered to be a strong contender as a unified theory of cortical function (Friston, 2005; Friston et al., 2006; 2012; Clark, 2015a; Seth, 2014). Moreover, as we will explore in more detail shortly, predictive coding has been shown to converge, under a variety of circumstances, to the BP of error algorithm used to train artificial neural networks, and thus may to provide a close link between modern machine learning techniques and biologically plausible theories of cortical function in the brain (Whittington & Bogacz, 2017; Millidge et al., 2020a; Song et al., 2020).

The Gaussian generative model of predictive coding also provides an additional set of *precision* parameters which correspond to the inverse variance parameters of the Gaussian distribution which we denote $\Pi$. These precisions modulate the strength of the prediction errors in the dynamics of the predictive coding network (Equations 19) such that we can simply define the 'precision-weighted prediction errors' $\tilde{\epsilon} \equiv \Pi\epsilon$ and otherwise follow the same dynamics.

### A.1.2 BACKPROPAGATION (BP)

The fundamental algorithm which has underpinned the recent major advances in machine learning has been the backpropagation of error algorithm. Arising in the 1970s (Linnainmaa, 1970) and popularised in the mid 1980s (Rumelhart & Zipser, 1985; Rumelhart et al., 1986), backpropagation was immediately applied to train connectionist artificial neural networks with some successes. However, only recently, with vastly increased computational power, and the harnessing of the innate parallelism of GPU units to train deep neural networks, has the true power and scalability of this algorithm become apparent (Goodfellow et al., 2016; Silver et al., 2016; Vaswani et al., 2017; Krizhevsky et al., 2012).

Backpropagation of error really refers to a more general suite of algorithms known as *Automatic Differentiation* (AD) (Griewank et al., 1989; Baydin et al., 2017), which take advantage of the chain-rule of calculus to allow for the computation of the gradients of any complex function made up of many differentiable subcomponents with numerical precision and with a cost that is linear in that of the original function. The key idea is that given a complex function $F$ which is the composition of many differentiable component functions $f$ such that $F \equiv f_1 \circ f_2 \circ \dots f_N$, and given some inputs $x$, we may compute $\frac{\partial F}{\partial x}$ by the chain rule as,

$$\frac{\partial F}{\partial x} = \frac{\partial f_N}{\partial f_{N-1}} \frac{\partial f_{N-1}}{\partial f_{N-2}} \dots \frac{\partial f_1}{\partial x} \tag{21}$$

This can apply not only to a straightforward 'chain', but to an arbitrary directed acyclic graph (DAG) of function composition. The generality of this approach comes from the fact that very many complex computations can be broken down in just this manner into a *computation graph* composed of differentiable functions. Thus, if we know the derivatives $\frac{\partial f_n}{\partial f_{n-1}}$ of every component function, then we can straightforwardly compute the derivative of the complex function in an extremely efficient and accurate way. The backpropagation of error algorithm performs *reverse-mode* AD in that it evaluates the chain of derivatives in Equation 21 recursively from the left – i.e. from the output of the network to the input. This can be implemented by accumulating gradients onto an *adjoint* vector $\delta$ which is initialized as $\delta_N \equiv \frac{\partial f_N}{\partial f_{N-1}}$ and then is updated according to the recursion,

$$\delta_{n-1} = \delta_n \frac{\partial f_n}{\partial f_{n-1}} \tag{22}$$

We can apply backpropagation explicitly to a standard feedforward neural network used for supervised classification to derive the necessary update rules. The network is composed of multiple layers, where the activity at each layer $x_l$ is a function of the activity of the layer below $x_l = f(W_{l-1}x_{l-1})$. At the output layer $x_L$, we are given a target $T$ and we compute how well the output matched the target using a loss function $\mathcal{L}(x_L, T)$. We then train the network to minimize this loss by computing the gradient of the loss with respect to the weight matrix for every layer $\frac{\partial \mathcal{L}}{\partial W_l}$, and then updating each weight matrix according to this gradient $W_l^{t+1} = W_l^t + \eta \frac{\partial \mathcal{L}}{\partial W_l}$ where $\eta$ is a scalar learning rate hyperparameter. To compute this gradient, we can apply backpropagation to the computational graph of the neural network. Typically, we first run the network 'forward' to compute the output $x_L$ and compute the loss $\mathcal{L}(x_L, T)$. Then, we begin the 'backward pass' which is where we compute the

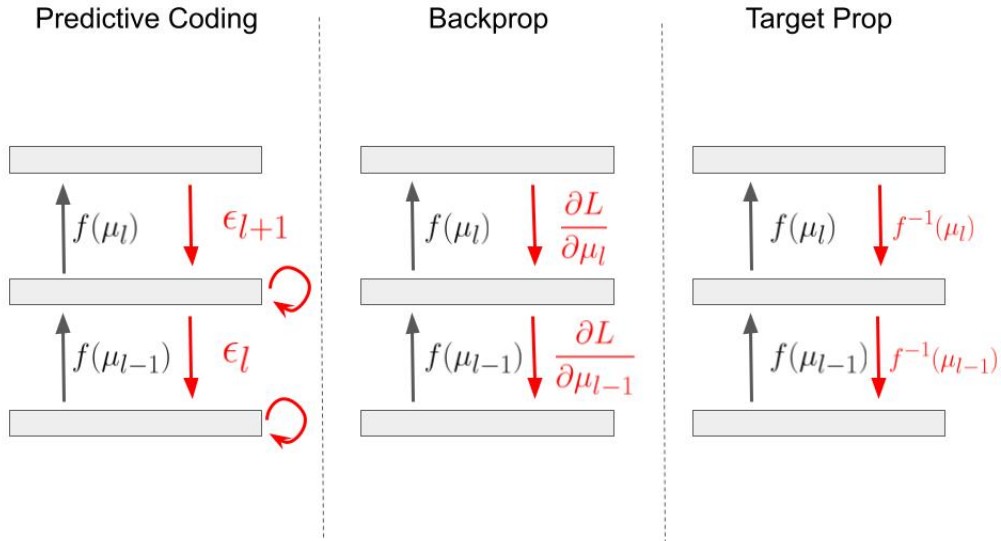

Figure 5: Comparison of learning rules for predictive coding, backpropagation, and target propagation. Predictive coding first proceeds with an iterative error minimization step before weight update based upon prediction errors between layers. Backpropagation propagates gradients backwards in its backwards phase and then uses the gradients to directly update weights. Targetpropagation propagates targets backwards in the form of layerwise inverses, and then uses the differences between target and activation to update weights. We will later see that the inference phase of predictive coding interpolates between backpropagation and target propagation.

adjoints using the backprop recursion in Equation 22. Specifically, we initialize the adjoint vector as $\delta_L = \frac{\partial \mathcal{L}}{\partial x_L}$ and then recursively compute the adjoints of the layer below using the relationship,

$$\delta_{l-1} = \delta_l f'(W_l x_l) W_l^T \tag{23}$$

Given the adjoints for each layer, we can then compute the weight gradient as $\frac{\partial \mathcal{L}}{\partial W_L} = \delta_l f'(W_l x_l) x_l^T$. While simple and extremely effective, the idea of the brain directly implementing and learning through the backpropagation of error algorithm has been criticized as biologically implausible (Crick, 1989).

### A.1.3  TARGET PROPAGATION

Target propagation is an alternative algorithm for training deep neural networks which was constructed to be more biologically plausible than backpropagation (Lee et al., 2015; Bengio, 2020; Bengio & Fischer, 2015). The core intuition behind targetpropagation (or targetprop) is that what we really want the activity at a layer to be *the activity that, when passed through the forward pass of the network, outputs the desired label or target exactly*. If we can compute this quantity – which we call a local *target* – then we can update the feedforward weights to minimize the difference between the actual activity and the local target, and thus ensure that eventually the feedforward pass of the network will compute the correct targets. targetpropagation thus relies on a local error signal involving the local targets which is argued to be more biologically plausible.

If we consider an output to be the composition of a set of functions, for instance $x_4 \equiv f_3(f_2(f_1(x_0)))$, then we can compute the optimal local target $T_0$ from the global target $T_4$ as the *inverse* of the functions $T_0 = f_1^{-1}(f_2^{-1}(f_3^{-1}(T_4)))$. From this, we can obtain a recursive relationship for the local targets, given a feedforward ANN structure ($x_l = f(W_{l-1}x_{l-1})$), which is simply,

$$T_l = f^{-1}(W_l^{-1} T_{l+1}) \tag{24}$$

Or, in the linear case $T_l = W^{-1} T_{l+1}$. Crucially, this requires computing the inverse of the weight matrix, which is computationally expensive and, more importantly, biologically implausible to

compute directly. Instead, the targetprop algorithms in the literature amortize this computation by utilizing a separate learnable set of backwards weights $\tilde{W}$ which are trained to minimize an autoencoder objective and thus ultimately learn to become an approximate inverse [1]. If we consider the linear case for simplicity, we can write out the key targetprop equations as,

$$T_l = \tilde{W}_l T_{l+1}$$
$$\dot{\tilde{W}}_l = -\tilde{e} x_l^T$$
$$\tilde{e} = x_l - \tilde{W} W x_l$$
$$\dot{W}_l = -e x_l^T$$
$$e = x_l - T_l \tag{25}$$

With the ideal case being where $\tilde{W} = W^{-1}$. In practice, learning the backwards weights often reduces in instabilities due to imperfect inverses and it has been found necessary to add a 'correction' term to the update rules accounting for the reconstruction loss of the implicit autoencoder formed by the backwards weights – a correction known as differential targetprop (Lee et al., 2015).

The empirical performance of targetpropagation in training artificial neural networks has been evaluated in a number of papers (Lee et al., 2015; Bengio, 2020; Meulemans et al., 2020), and recently there has been a significant theoretical advance in linking the procedure carried out by targetprop to the Gauss-Newton optimization method (Meulemans et al., 2020; Bengio, 2020) in the case where the target is simply a perturbation of the feedforward pass $- T_L = x_L + \beta \frac{\partial \mathcal{L}}{\partial x_L}$ where $\beta$ is taken to be a small scalar value. Since this perturbation is small, this allows for the network to be linearised around the initial feedforward pass values, and thus the computation of the targets implicitly computes a stack of inverse jacobians thus approximately implementing Gauss-Newton optimization.

Importantly, in this paper, we demonstrate a close and previously unknown relationship between predictive coding and target propagation. Specifically, we show that when the input layer of the predictive coding network, the equilibrium activity of the value neurons precisely equals the targetprop targets, and thus that predictive coding in this case can be thought of as iteratively computing the targetprop targets in a way that eschews the necessity to learn a set of backwards weights to compute the approximate inverse. More generally, we can interpret the equilibrium activities of predictive coding networks as being a balance between the feedforward pass values and the targetprop targets, with the ultimate goal of the minimization of the prediction errors being to set the feedforward pass activities equal to the targetprop targets (this being achieved when the prediction error is precisely 0). Moreover, we show that depending on the relative precision between layers, the equilibrium activity values after inference can be seen as an interpolation between their forward pass values and the targetprop targets, and thus that the weight updates in predictive coding networks interpolate between those predicted by backprop and by targetprop.

## A.2   PREDICTIVE CODING AND BACKPROP

While backpropagation has often been dismissed as biologically implausible, a set of recent works has demonstrated how predictive coding can converge to or approximate backprop weight updates in certain conditions. This means that predictive coding networks can be interpreted as performing or approximating the backproapgation of error algorithm in a potentially biologically plausible way. Since predictive coding is generally considered more biologically plausible, and indeed is often thought to be potentially implemented throughout the cortex (Clark, 2015b; Walsh et al., 2020; Bastos et al., 2012), then these equivalences may be extremely important in understanding the brain's ability to perform credit assignment.

Firstly, Millidge et al. (2020a) showed that if we keep the feedforward predictions 'frozen' to their feedforward pass values even as we update the activity values $x_l$ in accordance with Equation 19, then we can show that at the equilibrium of the network activities, the prediction error units $\epsilon$ satisfy the same recursive relationship as the backprop adjoints (Equation 22) and thus that if the final prediction error is set to the initial gradient of the loss function $\epsilon_L = \frac{\partial \mathcal{L}}{\partial x_L}$ then the prediction errors converge to

---

[1]It is also possible to design algorithms that compute these local targets online. Methods that do this in the literature are known as 'deep feedback control' (Podlaski & Machens, 2020; Meulemans et al., 2021).

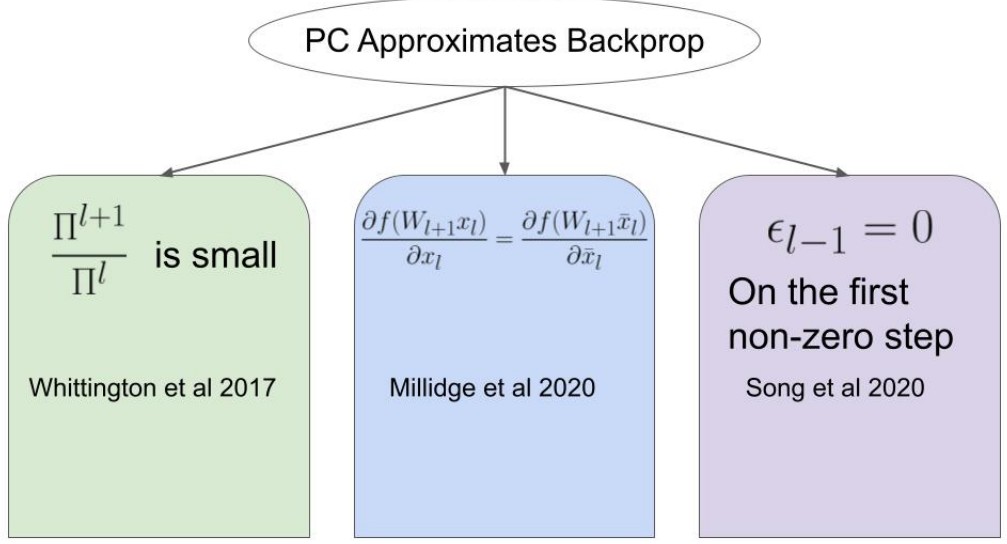

Figure 6: Summary of the results showing when predictive coding approximates or equals backprop. Firstly, when the precision ratio between layers is small (so the lower layers are heavily up-weighted) (Whittington & Bogacz, 2017). Secondly, under the fixed-prediction assumption, so that the optimization of each layer is decoupled (Millidge et al., 2020a). Thirdly, on the first update for a layer when the prediction error at the layer below is $0$ (Song et al., 2020).

the backprop gradients and thus the weight updates in the predictive coding scheme are identical to backprop. This approach, however, requires the relatively restrictive fixed-prediction assumption, which essentially prevents the convergence at one layer from influencing the convergence of any other layer, since the predictions between them are frozen at their initial feedforward pass values.

The key step is to notice that at the equilibrium, a recursive relationship for the prediction errors can be derived,

$$\dot{x}_l = 0 \implies \epsilon_l^* = \epsilon_{l+1} \frac{\partial f(W_{l+1}x_l)}{\partial x_l} \tag{26}$$

Crucially, if we assume that this partial derivative $\frac{\partial f(W_{l+1}x_l)}{\partial x_l}$ is taken at the feedforward pass values, instead of the current activity values (the frozen feedforward pass assumption), then this is exactly equivalent to the recursive relationship between the adjoints $\delta$ in backprop. Thus, if $\epsilon_L = \delta_L = \frac{\partial \mathcal{L}}{\partial x_L}$ then the prediction errors (and thus the ultimate weight updates) converge to those computed by backprop for any neural network or, indeed, any arbitrary computational graph.

Secondly, Song et al. (2020) recently showed that even if we relax the fixed prediction assumption, the *first* update of the activity values also leads to prediction errors precisely equalling the backprop gradients if the activity neurons $x$ are initialized to the feedforward pass values $x_l = f(W_l x_{l-1})$. This occurs because the prediction errors of the predictive coding network always satisfy the backprop recursion, and so converge to backprop whenever the activities are equal to the initial feedforward pass. This will always occur at the first step of the iteration as long as the activities are initialized to the feedforward pass values.

Finally, Whittington & Bogacz (2017) investigated the impact of precision (inverse-variance) parameters on the performance of predictive coding networks and demonstrated that when the precision of the output layer was very small, so that the activities at equilibrium were very close to their feedforward pass values, the prediction errors converged to the backprop gradients. In this paper we provide a general analysis of this phenomenon and show that it emerges when the precision ratio between feedback and feedforward precision is very low – i.e. that the feedforward path is weighted much more strongly than the feedback one. In such a scenario, the equilibrium activities converge to approximately the feedforward pass values, and so the prediction errors converge to backprop.

### A.3 BIOLOGICAL PLAUSIBILITY OF PCN LEARNING

### A.4 CONVEXITY IN THE LINEAR REGIME

It is straightforward to demonstrate that, in the linear case, the objective for a single layer (and thus the global objective which is just a sum of the local objectives) is convex and thus the global optimum can be easily located. This means that in the linear case there are no local minima and convergence is straightforward and rapid, which is indeed what we observe empirically.

The convexity can be straightforwardly shown by taking the second derivative of the free energy as follows,

$$\frac{\partial^2 \mathcal{F}}{\partial x_l^2} = I + W_{l+1}^T W_{l+1} \tag{27}$$

And then noting that $I + W_{l+1}^T W_{l+1}$ is positive semi-definite since,

$$z^T (I + W_{l+1}^T W_{l+1}) z = z^T z + z^T W_{l+1}^T W_{l+1} z = ||z||^2 + ||W_{l+1} z||^2 \geq 0 \tag{28}$$

for any arbitrary vector $z$. The fact that the second derivative of the objective function is always positive (indeed constant here) is sufficient to prove convexity.

### A.5 PATH TO CONVERGENCE

Since the dynamics are linear, we can also explicitly solve for the activities during convergence as a function of time as a linear differential equation. There is a slight complication that not only the current layer is changing during convergence but also the layer above and layer below which affect the dynamics of the current layer. This difficulty prevents an analytical solution to the dynamics for the general case (even when linear), but we can solve for the condition where the layers above and below can be treated as fixed. This can occur in reality either if they are fixed to the data (or label) or else because the other layers are considered to be already at equilibrium.

In this case, we can write out the dynamics as,

$$\begin{aligned} \dot{x}_l &= x_l - W_l x_{l-1} - W_{l+1}^T x_{l+1} + W_{l+1}^T W_{l+1} x_l \\ &= (I + W_{l+1}^T W_{l+1}) x_l - W_l x_{l-1} - W_{l+1}^T x_{l+1} \end{aligned} \tag{29}$$

Since, if we assume the endpoints are fixed, then we can take $-W_l x_{l-1} - W_{l+1}^T x_{l+1}$ to be constant with respect to time. In this case, the dynamics for $x_l$ are simply a linear differential equation of the form $\dot{x} = Ax + b$ where $A = (I + W_{l+1}^T W_{l+1})$ and $b = -W_l x_{l-1} - W_{l+1}^T x_{l+1}$. Equations of this canonical form can be solved analytically with the solution for time index $\tau$,

$$x_l(\tau) = e^{(I+W_{l+1}^T W_{l+1})t} x_l(0) - (e^{(I+W_{l+1}^T W_{l+1})t} - I)(I + W_{l+1}^T W_{l+1})^{-1} [W_l x_{l-1} + W_{l+1}^T x_{l+1}] \tag{30}$$

If we do not assume that the endpoints are fixed (as in general multilayer PCNs), we instead get a non-autonomous linear differential equation, with the canonical form $\dot{x} = Ax(\tau) + b(\tau)$, which has the following solution,

$$x_l(\tau) = e^{(I+W_{l+1}^T W_{l+1})t} x_l(0) + \int ds - e^{(I+W_{l+1}^T W_{l+1})(\tau-s)} [W_l x_{l-1}(\tau) + W_{l+1}^T x_{l+1}(\tau)] \tag{31}$$

Unfortunately in this case, since the feedback and feedforward components are themselves functions of time, this integral cannot be solved analytically and so instead must be approached numerically.

### A.6 PROOFS FOR INFERENCE RESULTS

#### A.6.1 PROOF OF THEOREM 3.5

Recall that in the linear case, the activities of a layer have the following dynamics,

$$\dot{x}_l = \Pi_l \epsilon_l - \Pi_{l+1} \epsilon_{l+1} W_l^T \tag{32}$$

By some somewhat involved algebraic manipulations we can explicitly solve for the fixed point of these dynamics to obtain an analytical expression for the precision-weighted equilibrium activities

$$
\begin{aligned}
\dot{\mu}_l = 0 \implies & \Pi_l \epsilon_l - \Pi_{l+1} \epsilon_{l+1} W_l^T = 0 \\
\implies & \Pi_l \mu_l - \Pi_l W_{l-1} \mu_{l-1} - W_l^T \Pi_{l+1} \mu_{l+1} W_l^T \Pi_{l+1} W_l \mu_l = 0 \\
\implies & \mu_l - W_{l-1} \mu_{l-1} - \Pi_l^{-1} W_l^T \Pi_{l+1} \mu_{l+1} \Pi_l^{-1} W_l^T \Pi_{l+1} W_l \mu_l = 0 \\
\implies & (I + \Pi_l^{-1} W_l^T \Pi_{l+1} W_l) \mu_l - W_{l-1} \mu_{l-1} - \Pi_l^{-1} W_l^T \Pi_{l+1} \mu_{l+1} = 0 \\
\implies & \mu_l^* = (I + \Pi_l^{-1} W_l^T \Pi_{l+1} W_l)^{-1} \left[ W_{l-1} \mu_{l-1} + \Pi_l^{-1} W_l^T \Pi_{l+1} \mu_{l+1} \right]
\end{aligned}
\tag{33}
$$

Where we can observe the previous activity equilibrium in Theorem 3.3 as a special case where $\Pi_l = \Pi_{l+1} = I$. Precision matrices are always square so in this case exact inverses are plausible.

### A.6.2 Proof of equal feedforward and feedback input at $0$ prediction error

By a straightforward algebraic manipulation of the dynamics in the linear case, we can determine the necessary condition for the prediction error at a layer to be $0$,

$$
\begin{aligned}
\epsilon_l^* = 0 \implies & x_l^* - W_l x_{l-1}^* = 0 \\
\implies & \left[ W_l x_{l-1}^* + W_{l+1}^T x_{l+1}^* \right] - (I + W_{l+1}^T W_{l+1}) W_l x_{l-1}^* = 0 \\
\implies & W_{l+1}^T x_{l+1}^* - W_{l+1}^T W_{l+1} W_l x_{l-1}^* = 0 \\
\implies & W_{l+1}^T W_{l+1} W_{l-1} x_{l-1}^* = W_{l+1}^T x_{l+1}^* \\
\implies & W_l x_{l-1}^* = W_{l+1}^{-1} x_{l+1}^*
\end{aligned}
\tag{34}
$$

which occurs when the feedforward and feedback influences are exactly equal or, alternatively, when the feedforward prediction at a layer equals the targetprop target. Thus, the global minimum, where $\epsilon_l^* = 0$ for all layers, is attained when throughout the network the feedforward pass value exactly equals the targetprop target or, alternatively, when the feedforward pass value exactly predicts the correct output target. This provides an additional intuitive reason why PCNs converge in practice, which is that their global minimum is achieved at $0$ training error.

### A.6.3 Dynamical Proof of Input-Unconstrained Network

Here we provide an alternative proof of the fact that an input-unconstrained PCN will converge to the targetprop targets. The input being unconstrained means that for each layer, the layer below can, in theory, perfectly track the activity of that layer, meaning that we can consider $\epsilon_l = 0$. This means that the dynamics for the activities need only consider the feedback prediction error and can be written as,

$$
\begin{aligned}
\dot{x}_l &= -\epsilon_{l+1} \frac{\partial f(W_{l+1} x_l)}{x_l} \\
&= (x_{l+1} - f(W_{l+1} x_l)) f'(W_{l+1} x_l) W_{l+1}^T
\end{aligned}
$$

From this, we can directly solve for the equilibrium as follows,

$$
\begin{aligned}
\dot{x}_l = 0 \implies & (x_{l+1} - f(W_{l+1} x_l)) f'(W_{l+1} x_l) W_{l+1}^T = 0 \\
\implies & x_{l+1} - f(W_{l+1} x_l) = 0 \\
\implies & x_l^* = W_{l+1}^{-1} f^{-1}(x_{l+1})
\end{aligned}
\tag{35}
$$

Which is precisely the same recursion as that satisfied by the targetprop targets in Equation 24

### A.7 Proofs for Learning Results

### A.7.1 Proof of Prosposition 3.1

Recall that $\mathcal{F} = L + \tilde{E}$. Thus, if $\mathcal{F} = 0$ then we have,

$$
0 = L + \tilde{E}
\tag{36}
$$

Now, $\tilde{E}$ is defined as a sum of layerwise energies which are themselves (in standard PCNs) defined as a squared prediction error,

$$\tilde{E} = \sum_l^{L-1} E_l \tag{37}$$

$$E_l = \sum_i^N \epsilon_{l,i}^2 \tag{38}$$

Since the squared prediction errors cannot be negative, $\tilde{E}$ is non-negative. Thus, if we assume that $\mathcal{L}$ also cannot be negative (a standard, but not always fulfilled property of a loss function), then we can obtain the desired result.

### A.7.2 Proof of Marginal Lemma

Recall the decomposition of the free energy into the backprop loss and the residual error $\mathcal{F} = L + \tilde{E}$. If we differentiate this expression with respect to $x_l$, we obtain,

$$\frac{\partial \mathcal{F}}{\partial x_l} = \frac{\partial \mathcal{L}}{\partial x_l} + \frac{\partial \tilde{E}}{\partial x_l} \tag{39}$$

Now, at equilibrium we have that,

$$\dot{x}_l = -\frac{\partial \mathcal{F}}{\partial x_l} = 0 \tag{40}$$

Thus,

$$\frac{\partial \mathcal{F}}{\partial x_l} = \frac{\partial \mathcal{L}}{\partial x_l} + \frac{\partial \tilde{E}}{\partial x_l} = 0 \tag{41}$$

$$\implies \frac{\partial \mathcal{L}}{\partial x_l} = -\frac{\partial \tilde{E}}{\partial x_l} \tag{42}$$

which is simply the marginal condition.

### A.7.3 Derivation of energy gradient bound

The fundamental condition we wish to preserve is that the supervised loss can never increase during inference, and we want to express this condition in terms of the balance of supervised and internal energies. We can express this condition as,

$$\frac{d\mathcal{L}}{dt} = \frac{\partial \mathcal{L}}{\partial x_l}^T \frac{dx_l}{dt} \leq 0 \tag{43}$$

for every $x_l$. Using the fact that the dynamics of the EBM are gradients of the total energy, we can rewrite this as,

$$\frac{\partial \mathcal{L}}{\partial x_l}^T \frac{dx_l}{dt} = -\frac{\partial \mathcal{L}}{\partial x_l}^T \frac{\partial \mathcal{F}}{\partial x_l} \leq 0 \tag{44}$$

$$= -\frac{\partial \mathcal{L}}{\partial x_l}^T \frac{\partial \tilde{E}}{\partial x_l} - \frac{\partial \mathcal{L}}{\partial x_l}^T \frac{\partial \mathcal{L}}{\partial x_l} \leq 0 \tag{45}$$

$$\implies -\frac{\partial \mathcal{L}}{\partial x_l}^T \frac{\partial \tilde{E}}{\partial x_l} \leq ||\frac{\partial \mathcal{L}}{\partial x_l}||^2 \tag{46}$$

which is exactly the condition presented in the main text. Also note that with equality this condition reduces to the margnial condition, since the marginal condition obtains at equilibrium when the change of the loss must be $0$.

Intuitively, we can think of the energy gradient bound as requiring that the dot product similarity between the supervised residual loss be at least as large as the negative of the dot product similarity between the backprop loss and itself. In essence, this places a bound on the degree of negative correlation possible between the supervised and residual loss so that the residual loss cannot 'overpower' the supervised loss, although far away from inference both can become very large if they are correlated with each other so that a movement may induce a large decrease in both losses at the same time.

Also of interest is that when the network is initialized to its forward pass values, so $\frac{\partial \tilde{E}}{\partial x_l} = 0$, then any supervised loss gradient is guaranteed to satisfy the bound.

### A.8 RELATIONS BETWEEN PREDICTIVE CODING, TARGET PROPAGATION, AND BACKPROPAGATION

Returning to the missing data intuition behind the interpretation of predictive coding as an EM algorithm, we can think about the inference phase of the PC network as performing a *constrained* inference of the activity values or, alternatively, performing Bayesian inference with priors. The inference consist of trying to infer the optimal local targets while also staying close to their feedforward pass initializations. However, what happens if we perform an unconstrained minimization during the inference phase? If this is the case, then we will infer the optimal local targets, and hence the PC network will converge towards performing target propagation. This intuition is verified by our experiments in Figure 1 showing that when PC networks are input unconstrained (i.e. there is no prior keeping them close to the feedforward pass values) the equilibrium converges towards the optimal targetprop targets. Unconstrained predictive coding, then, is effectively an iterative form of target propagation where the targets are inferred dynamically for each batch whereas in classical target propagation the targets are inferred using a fixed set of backwards weights, which are learnt across a dataset, thus instead inferring the targets through amortized inference (Marino et al., 2018; Welling & Teh, 2011; Tschantz et al., 2022).

Another way to think about this is to return to the decomposition of the free energy into $\mathcal{L}$ and $\tilde{E}$. We can see that the magnitude of $\mathcal{L}$ controls the strength of the likelihood term driving the activities at equilibrium towards the targets while the magnitude of $\tilde{E}$ controls the strength of the prior term keeping the activities constrained to their feedforward pass values. If we postulated a parameter $0 \leq \lambda \leq 1$ that controls this trade-off, such that we could write the free energy $\mathcal{F}$ as,

$$\mathcal{F} = \lambda L + (1 - \lambda)\tilde{E} \tag{47}$$

We could see that setting $\lambda = 1$ results in target propagation since the inference procedure is unconstrained. Alternatively, by setting $\lambda = 0$, we obtain pure backpropagation, since the inference phase becomes trivial as the activities are constrained to exactly equal their feedforwad pass values. We can then understand predictive coding as being an interpolating solution with $0 < \lambda < 1$ between target propagation and backpropagation where $\lambda$ is implicitly set by the ratio of precisions between the output layer and the rest of the network layers. This result is shown graphically in Figure 2A.

Putting these aspects together also allows us to understand all the previous equivalences between backpropagation and predictive coding presented in the literature. Whittington & Bogacz (2017) found that predictive coding approximated backprop when the ratio of precision was strongest at the input layers of the network, decreasing exponentially towards the output. We can now understand this in terms of effectively setting $\lambda \approx 0$ such that the $\tilde{E}$ term becomes dominant, the equilibrium activities are very close to the feedforward pass activities, meaning that $\mathcal{L}_{PC} \approx \mathcal{L}_{BP}$ where we can define $\mathcal{L}_{BP}$ as the backprop loss through the feedforward pass activity values and $\mathcal{L}_{PC}$ as the backprop loss on the equilibrium activity values, and so the gradients computed by predictive coding approximate backprop. Millidge et al. (2020a), on the other hand, show that predictive coding converges to exactly backprop under the 'fixed prediction assumption'. This assumption essentially fixes the layerwise gradient $\frac{\partial f(W_l x_l^*)}{\partial x_l^*}$ to the feedforward pass values $\frac{\partial f(W_l x_l^*)}{\partial x_l}$ and thus directly enforcing convergence to the exact backprop gradients.

Finally Song et al. (2020) note that if the activity values are initialized to their feedforward pass values, then the initial prediction errors for each layer except the output are 0, then for each inference step, errors propagate one layer backwards through the network. They note that after the *first* inference step for each layer, when the error first becomes nonzero, the error is exactly equal to the backprop gradients. Again this is because at the first step of inference, the activities of the layer are precisely equal to the feedforward pass values. A simple way to understand this is to recall the decomposition of the free energy into the residual and the backprop loss (Equation 11: $\mathcal{F} = \tilde{E} + \mathcal{L}$, noting that when the network is initialized to its feedforward pass values then the network is at a global minimum with respect to the activities – i.e., $\tilde{E} = 0$ and $\frac{\partial \tilde{E}}{\partial x} = 0$. If we differentiate through the free energy decomposition with respect to the activities $x$, we obtain,

$$\frac{\partial \mathcal{F}}{\partial x} = \frac{\partial \tilde{E}}{\partial x} + \frac{\partial \mathcal{L}}{\partial x}$$

$$\frac{\partial \tilde{E}}{\partial x} = 0 \implies \frac{\partial \mathcal{F}}{\partial x} = \frac{\partial \mathcal{L}}{\partial x}$$

Thus, the activity gradients at the first update away from the feedforward pass will be exactly in the direction of the gradient. Since we then use the activity gradients to directly compute the weight gradients, the weight updates on the first step are the exact backprop gradients. Note that this derivation applies to any energy based model with an additive decomposition of the energy as in Equation 11 and is not specific to predictive coding.

Lastly, our theoretical framework also provides an explanation for the fact that in practice predictive coding networks trained with redistribution work best when they are initialized to the feedforward pass values of the corresponding backprop ANN before the inference phase begins. This 'trick' has been widely, albeit mostly implicitly, used in the literature (Song et al., 2020; Salvatori et al., 2021; Millidge et al., 2020a;b) and it has been observed that predictive coding networks perform poorly or do not work at all without it. We can now see that the reason for this relates to the condition for the constrained minimization interpretation of the inference process needing to be initialized on the 'correct side' of the marginal condition – i.e. $\frac{\partial \mathcal{L}}{\partial x} \geq \frac{\partial \tilde{E}}{\partial x}$. If, through a random initialization, the network starts out on the 'opposite side'; i.e, $\frac{\partial \mathcal{L}}{\partial x} \leq \frac{\partial \tilde{E}}{\partial x}$, then the inference procedure, by convergence to the marginal condition will tend to decrease $\tilde{E}$ by *increasing* $\mathcal{L}$, thus violating the conditions for the convergence of the E-M algorithm. The reason the network appears to work best at the feedforward pass initialization is that here the network starts out with 0 prediction errors except at the output, meaning that $\tilde{E} = 0$ at initialization. This is therefore the situation in which the network begins 'furthest away' from the marginal condition, and can therefore achieve a large decrease in $\mathcal{L}$ before the corresponding increases in $\tilde{E}$ force the network to an equilibrium.

### A.9 GENERALIZING PC TO OTHER ENERGY FUNCTIONS

As a consequence of these theoretical results, it is also possible to investigate whether the energy function of predictive coding can be generalized to give a family of possible energies and associated network architectures which all possess the favourable inference and learning properties of predictive coding networks and which may even lead to more effective inference or learning. Retracing the steps of our argument, several key conditions are apparent. First, the total energy $\mathcal{F}$ must be bounded below, ideally with a minimum at 0. Secondly, the total energy must be expressable as a sum of the backprop loss and some residual energy $\tilde{E}$ such that the marginal condition can be derived. Finally, for the weight updates to be equivalent to backprop through the equilibrium network, the prediction errors must satisfy the adjoint recursion in backprop (Equation 22).

Although this condition of satisfying the adjoint recursion appears quite stringent, it actually allows quite a degree of generalization. Suppose that our residual energy function $\tilde{E}$ is still composed of layerwise energies which are composed of an 'interaction function' $g(x_l, x_{l-1})$ of the activities of the layer and the layer below, as well as an 'activation function' $h(g)$ which is a function of the result of the interaction function. This lets us write the residual energy as,

$$\tilde{E} = \sum_l h(g(x_l, x_{l-1})) \tag{48}$$

From this generalized formulation we can recover predictive coding by setting $g = (x_l - f(W_l, x_{l-1})$ and $h(x) = x^2$. With this definition of the energy, we can derive the dynamics of a layer,

$$\dot{x}_l = \frac{\partial h_{x_l}}{\partial g} \frac{\partial g(x_l, x_{l-1})}{\partial x_l} + \frac{\partial h_{x_{l+1}}}{\partial g} \frac{\partial g(x_{l+1}, \partial x_l)}{\partial x_l} \tag{49}$$

Where, for ease of notation, we shorten $\frac{\partial h(g(x_l, x_{l-1}))}{\partial g(x_l, x_{l-1})}$ to $\frac{\partial h_{x_l}}{\partial g}$. From these dynamics, it is clear we can recover the adjoint recursion if we set,

$$\frac{\partial g(x_{l+1}^*, x_l^*)}{\partial x_l^*} \div \frac{\partial g(x_l^*, x_{l-1}^*)}{\partial x_l^*} = -\frac{\partial f(W_l x_l^*)}{\partial x_l^*} \tag{50}$$

This condition essentially means that the ratio of the gradients of the interaction functions between the current layer and the layer below and the current layer and the layer above are equal to the gradients of the feedforward pass of the layer $f(W_l x_l)$. This condition holds for predictive coding because $\frac{\partial g(x_l^*, x_{l-1}^*)}{\partial x_l^*} = 1$ and $\frac{\partial g(x_{l+1}^*, x_l^*)}{\partial x_l^*} = \frac{\partial (x_{l+1}^* - f(W_l x_l^*))}{\partial x_l^*} = -\frac{\partial f(W_l x_l^*)}{\partial x_l^*}$. Of note is that the adjoint recursion functions for any 'activation function' $h$ so long as the derivative ratio condition is

met. This is because, no matter what they are, the activation function gradients $\frac{\partial h_{x_L}}{\partial g}$ take the form of the adjoint terms. This immediately leads to many possible generalizations. One potentially useful one, which has already been independently discovered in the literature (Alonso & Neftci, 2021) is to set $h(x) = \log(x)$. This gives rise to divisive prediction errors since $\frac{\partial h_{x_l}}{\partial g}$ becomes $\frac{1}{\epsilon_l}$.

Another, more principled, method of generalization is to return to the Bayesian interpretation of the inference objective derived in Equation 8. Notice that the inference objective is split into a likelihood term $p(y|x, D, W)$ and a prior term $p(x|D, W)$. Our decomposition of the free energy into the backprop loss and the residual loss allows us to associate the likelihood term with $\mathcal{L}$ and the prior term with $\tilde{E}$. This means that we can interpret the 'constraints' $\tilde{E}$ in a principled way as Bayesian priors. Specifically, we can make the identication,

$$\tilde{E} = -\ln p(x|D, W) = -\sum_l \ln p(x_l|x_{l-1}, W_l) \tag{51}$$

Which derives from the layerwise composition of $\tilde{E}$ or, equivalently, from the factorization of the prior. Given this understanding of the residual loss as a Bayesian prior, we can understand that in the case of predictive coding networks this prior is Gaussian around the predictions from the layer below. Effectively, it enforces a quadratic penalty to stop the newly inferred activities deviating too far from their initial conditions. The strength of this prior is controlled by the relative precision parameters in each layer. This leads to the suggestion that a natural generalization of predictive coding would be to utilize other prior distributions, perhaps ones which are more natural for specific problems, and which depend on the statistics of the data in question. For instance, if the data is strongly log-normal, or power-law distributed, it may be that using log-normal priors as well as divisive prediction errors may lead to better inductive biases for the network and ultimately more efficient inference and learning.

## A.10 METHODS

### A.10.1 INFERENCE RESULTS

For the numerical results on inference in predictive coding networks described in this paper we used an extremely simple 3-layer predictive coding scheme designed to showcase our mathematical results in a straightforward way.

All layers had five neurons and the input layer and output layer were fixed to the 'data' and the 'target' which in this case were randomly generated from Gaussian distributions with a mean of 1 and -1, respectively, and a standard deviation of 1. The middle-layer was free to vary according to the predictive coding update rules. For the nonlinear networks we used a hyperbolic tangent activation function. We typically ran inference for 100 timesteps using a learning rate of $0.05$. Weight matrices were randomly generated from a Gaussian distribution with a mean of 0 and a variance of $0.05$.

For the results on the nonlinear fixed point iteration algorithm we utilized a randomly initialized four layer perceptron with 20 dimensional random Gaussian inputs and outputs and a hidden width of 20 neurons. We initialized all weights from a zero-mean Gaussian with a variance of $0.05$.

For the precision experiments, we initialized mostly diagonal covariance matrices with a diagonal set to a certain value and the value of other entries sampled from a random gaussian with a mean of zero and a variance of $0.1$. To compute the precision matrices we then inverted the covariance matrices.

In the mathematical notation in the paper we freely use the matrix inverse $W^{-1}$. However, in practice, weight matrices are usually not square. In the numerical results we instead replace the true inverse with the pseudoinverse $W^{\dagger}$. Although this makes many of our mathematical results only approximate, in practice, our numerical simulations have lead us to believe that this approximation does not dramatically affect the intuitions conveyed in this paper, so long as the layer-wise width ratios do not become too extreme.

### A.10.2 LEARNING RESULTS

In the learning experiments, we utilized a three layer MLP trained on the MNIST and Fashion-MNIST datasets. The MNIST digits were normalized to lie between 0 and 1. The network had hidden layers of size 128 and 64. The first two layers had ReLU activation functions and the output layer was an identity. The network was trained using the mean squared error loss function using a Nesterov

momentum optimizer using a learning rate of $0.0001$ and a momentum value of $0.9$. In general, results are averaged over 5 seeds and the shaded area represents the standard deviation.

Code to reproduce all experiments and figures in this paper will be made available after review at `github_address`. It is also included in the supplementary material folder.

