# OpenReview forum: "A Theoretical Framework for Inference and Learning in Predictive Coding Networks"
_ICLR.cc/2023/Conference — ICLR 2023 poster_

### Official Review · Reviewer_c2Qw · 2022-10-23

**Confidence:** 4
**Clarity, Quality, Novelty And Reproducibility:** The paper is very clearly written, an…
**Correctness:** 4
**Technical Novelty And Significance:** 4
**Empirical Novelty And Significance:** Not applicable
**Recommendation:** 8

**Details Of Ethics Concerns:**

It would have not hurt the paper if they did not make so many self references. As it stands, this paper stretches what is permissible in a double blind review.

**Strength And Weaknesses:**

Strength: The analysis is quite comprehensive and connections to BP and TP are shown
Weekness: Several of the results are for linear networks where the problem becomes trivial. Also, a lot of the results are similar to deep gaussian networks that came out many years back.

**Summary Of The Paper:**

I would like to begin by saying that this paper makes repeated and relentless reference to a given set of papers that induced this reviewer to suspect that it was self reference, and it took barely a minute of google search to verify this fact. I find this very disturbing, since the other papers I have reviewed have by and large played by the rules of double blind.

That said, this is a very interesting paper. The authors consider a feedforward ANN, and analyze an algorithm called prospective configuration that falls within the perview of predictive coding. Basically, given an data point and its label, the input and output of the network are clamped to their respective input and output values, and each intermediate layer searches thru gradient descent for input values that would minimize the squared difference between its output and the current output. This allows the error at the top layer to propagate downwards. Once equilibiirm has been reached, the input outputs of each layer is clamped and a gradient descent on the weight vectos is initiated, again until convergence. This process is repeated until convergence.

What the paper shows is that this this algorithm converges to the same local optima that back propagation converges to. It also show that the algorithm builds a bridge between BP and target propagation (an alternative learning algorithm for deep networks).
It builds its results on the intuitive idea that the problem can be rephrased as a missing data problem, that is the activation of the intermediate layers) and can therefore be attacked using EM.

**Summary Of The Review:**

This is a well written paper that provides theoretical justification for yet another technique (prospective configuration) for training deep networks.

---

> ### Author Response · Authors · 2022-11-15
> **Response to reviewer c2Qw**
>
> We thank the reviewer for their detailed review. With regards to the self-reference, we did not realize it may be excessive and this was not intentional on our part. We believe that we have fairly cited all relevant literature and all citations are relevant to the work.
>
> As to the weaknesses of the paper, you highlight:
>
> 1.) While many of our arguments are for linear networks, this is because these are the only networks that can be solved analytically. We argue that the primary intuitions behind these results transfer to nonlinear models of interest and show empirical results to this effect. Also, some of our main results relating to the relationship between prospective configuration and target propagation, as well as the convergence of PC and BP apply also to nonlinear networks.
>
> 2.) We are not aware of the links the reviewer alludes to with deep gaussian networks. We would appreciate some citations to relevant work the reviewer has in mind so we can try to understand the potential connections and add it in to the related work

---

### Official Review · Reviewer_3s9Y · 2022-10-25

**Confidence:** 4
**Correctness:** 2
**Technical Novelty And Significance:** 2
**Empirical Novelty And Significance:** Not applicable
**Recommendation:** 3

**Clarity, Quality, Novelty And Reproducibility:**

While this is a bold attempt at some theory, several claims seem dubious and others are often trivial.

**Strength And Weaknesses:**

Strengths:
Understanding the relation between PCN and usual ANN with BP is definitely a worthy goal.
The observation that PCN ‘inference’ is a combination of a forward and a backward signal and that the balance between the two is decided by precision ratios. This insight may not be completely new, but is brought out well in the discussion.
Weaknesses:
The proofs of some theorems seem to be spoiled by what looks like sloppy or confusing arguments. Take for example theorem 3.1. In the proof, equation 3 ignores the difference between $\epsilon^*_{l+1}$, the equilibrium value, and $\epsilon_{l+1}$, the backward propagated error, to make the argument.
In some cases, it is not clear whether some conditions are non-trivial or not, if taken literally. For example, for theorem 3.5, if one thinks $\mathcal{L}$ depends only on $x_{L-1}$ and the target, the partial derivative $\partial \mathcal{L}/\partial x_{l}$ would naively be zero for $l<L-1$, making the inequality trivially true.
Calling the $x_l$ ‘inference’ via a local mode, not the expectation, the E-step Eq (14), and then invoking theorems about local convergence of expectation maximization seems confusing.

Minor issues:
In equation (2), it seems $\epsilon_l\cdot f’(..)$ is column vector of the same size as $x_l$. Given that, in equation (1), should the order of $W^T_{l+1}$ and $\epsilon_{l+1}\cdot f’(...)$ not be reversed?


**Summary Of The Paper:**

The paper attempts to provide a theoretical analysis computation in predictive coding networks (PCNs), relating it to the more commonly used backpropagation-based (BP) training of feedforward artificial neural nets (ANNs).

**Summary Of The Review:**

I wish this paper was what it promises to be. I believe far more careful work is necessary to substantiate the claims made here.

---

> ### Author Response · Authors · 2022-11-15
> **Response to reviewer 3s9Y**
>
> We thank the reviewer for their detailed and critical review. Overall we disagree with the assertion that our claims are dubious or trivial; we believe that this paper offers a state of the art and correct analysis of the learning and inference dynamics of PCNs including providing a full theoretical understanding of the limits in which PCNs converge to backprop, as well as a convergence proof of the prospective configuration algorithm, all of which are novel.
>
> The reviewer, however, has highlighted some sloppiness or mistakes in notation, and we are thankful for that. These have been fixed in the updated version. Specifically:
>
> In equation 3, we did indeed make a typo and the \epsilon_l should be \epsilon^*_l. This does not affect the validity of the proof since the whole point of the proof is taking the limit where \epsilon_f tends to 0 and the backprop recursion is identical whether $\epsilon$ or $\epsilon^*$ is used since for the output prediction error they are the same: $\epsilon = \epsilon^*$ and thus the rest of the recursion is identical. We have fixed this notational typo in the updated version.
>
> Secondly, the reviewer is correct that we the order of the weights and error terms in Equation 2 should be transposed and we have fixed this in the updated version.
>
> With regards to theorem 3.5, we do not assume that the loss function L depends only on x_{L-1} and hence the partial derivative is not trivial. In a feedforward architecture, L depends directly only on x_{L-1} and the penultimate layer weights, but x_{L-1} also depends on the previous activations in the network, and hence the partial derivatives are not 0.
>
> As for the reviewer’s comment:
>
> > Calling the xl ‘inference’ via a local mode, not the expectation, the E-step Eq (14), and then invoking theorems about local convergence of expectation maximization seems confusing.
>
> We do not quite understand the point the reviewer is trying to make here and would appreciate this being expanded upon. A key and explicit assumption that we make in predictive coding is that of a delta function approximation (alternatively Laplace approximation but it works out the same) around the variational posterior which is parametrized by x. With this assumption, the mode and expectation of the posterior are identical. This should not affect the probabilistic interpretation of the model as performing variational inference nor the results with expectation maximization.
>
> Overall, given that the reviewer has, in our eyes, only brought up relatively minor typos and misunderstandings, it does not support their claim that the results are dubious or trivial nor their low score. We would be especially interested in hearing in what ways the reviewer thinks our results, especially our convergence results and relationship to target-propagation are not novel, since we have not seen them before in the literature.

---

### Official Review · Reviewer_G2QQ · 2022-10-25

**Confidence:** 4
**Correctness:** 4
**Technical Novelty And Significance:** 4
**Empirical Novelty And Significance:** 4
**Recommendation:** 10

**Clarity, Quality, Novelty And Reproducibility:**

The present study builds on extant work that connects predictive coding and backpropagation, further elucidating the relationship of predictive coding to backprop and target prop away from prior limits studied.  The authors describe their analysis clearly and thoroughly, including demonstrating how results with full nonlinear networks are consistent with their linear network findings.  The proofs in the body of the paper are further supplemented with further descriptions and proofs in the appendices, while the numerical experiments are detailed in the appendix and will be included in a code release, allowing others to reproduce the results in a manner that is aided by, but not dependent on, the provided code.

**Strength And Weaknesses:**

Strengths

1. The authors provide a full mathematical characterization of linear predictive networks, demonstrating analytically their relationships to backprop and targetprop networks

2. Furthermore, simulated nonlinear networks are analyzed and shown to produce the same qualitative results as found for the linear networks

3. The authors' writing is clear and well-organized

Weaknesses

Major

No major weaknesses have been
observed.

Minor

The precision ratio is introduced after Theorem 3.4 as $\Pi_{l+1} / \Pi_l$; however, I believe the precision ratio comprising the x-axis values in Fig. 3C is instead $\Pi_{l} / \Pi_{l+1}$.  Either changing the figure to have the same ratio that is initially introduced or specifying the ratio that is used in the figure would help to avoid readers' confusion.

**Summary Of The Paper:**

While backpropagation is the mainstay technique used for artificial neural networks, it is considered biologically implausible, mainly due to the so-called "weight-transport" problem, wherein neurons need to know the weights of neurons that lie upstream in order to correctly update their own weights. Two frameworks that have been proposed to be more biologically realizable are predictive coding and target prop.  While both of these methods have been found to be related to backpropagation under certain circumstances, a fuller accounting of their interconnections has remained elusive. Through both mathematical analysis of linear networks and simulations of nonlinear ones, the present work demonstrates how backprop and targetprop approximately represent two endpoints of a continuum of solutions that predictive coding network inferences lie on (i.e., the equilibrium prediction errors lie close to the gradients of backprop on one end, while the equilibrium activities lie close to targetprop target activities on the other). Moreover, the authors show that predictive coding learning results in convergence of the network to the same minima as for backpropagation.

**Summary Of The Review:**

The authors have analytically demonstrated the connections among predictive coding, targetprop, and backprop in linear networks. They back up the analysis with numerical experiments on nonlinear networks that demonstrate the same qualitative results.  This work represents an important and novel advance in understanding predictive coding networks, which currently represent one main direction within neuroscience for understanding cortical functionality.

---

> ### Author Response · Authors · 2022-11-15
> **Response to reviewer G2QQ**
>
> We thank the reviewer for their enthusiastic and comprehensive review. About the precision ratio in Figure 3.C – this is a good spot and we have added a clarification of this to the figure caption.

---

### Official Review · Reviewer_3qX4 · 2022-10-25

**Confidence:** 2
**Correctness:** 3
**Technical Novelty And Significance:** 3
**Empirical Novelty And Significance:** 3
**Recommendation:** 8

**Clarity, Quality, Novelty And Reproducibility:**

- Quality and clarity
The math proof and derivations are sound and are presented clearly.

- Originality
The theoretical results seem novel but I am not fully sure since I am not an expert on PCN.

**Strength And Weaknesses:**

This is a paper full of math proof and derivations, and is theoretically sound. It is quite interesting in interpreting the PNC dynamics as performing a constrained EM algorithm, and comparing the PCN with other algorithms including target propagation and back propagation. The theoretical results seem novel based on the author's description but I am not quite sure since I am not an expert on predictive coding networks.

**Summary Of The Paper:**

The paper presents a novel theoretical framework for understanding the inference and the learning in predictive coding networks (PCN). The math derivations and proof are sound.

**Summary Of The Review:**

I like the deep math analysis of the inference and learning dynamics of PCN and it greatly gain our understanding of this network.

---

> ### Author Response · Authors · 2022-11-15
> **Response to reviewer 3qX4**
>
> We thank the reviewer for their detailed and positive review.

---

### Decision · Program_Chairs · 2023-01-20

**Decision:**

Accept: poster

**Justification For Why Not Higher Score:**

While the work appears to be solid, the proofs are only for linear networks and only empirical evidence is provided for nonlinear networks.

**Justification For Why Not Lower Score:**

NA

**Metareview: Summary, Strengths And Weaknesses:**

The paper presents a novel theoretical framework to further our understanding of predictive coding networks (PCN). The authors derive analytical results establishing connections between predictive coding, targetprop, and backprop in linear networks. They further provide experimental validation suggesting that the theory applies to nonlinear networks. Three of the four reviewers reported that the work was sound while a fourth reviewer noted some errors in the proofs (which the authors appeared to have addressed during the rebuttal but this was not confirmed by the reviewer who did not update their score/review).

**Note From Pc:**

if the above contains the word "oral" or "spotlight" please see: "oral" presentation means -> notable-top-5% and "spotlight" means -> notable-top-25%. As stated in our emails, we are disassociating presentation type from AC recommendations